# Virtual Gene Concept and a Corresponding Pragmatic Research Program in Genetical Data Science

**DOI:** 10.3390/e24010017

**Published:** 2021-12-23

**Authors:** Łukasz Huminiecki

**Affiliations:** Evolutionary, Computational, and Statistical Genetics, Department of Molecula Biology, Institute of Genetics and Animal Biotechnology, Polish Academy of Sciences, Postępu 36A, Jastrzębiec, 05-552 Warsaw, Poland; l.huminiecki@igbzpan.pl

**Keywords:** gene concept, technology, scientific method, experimentalism, molecular biology, genomics, bioinformatics, computational biology, data science, virtualization

## Abstract

Mendel proposed an experimentally verifiable paradigm of particle-based heredity that has been influential for over 150 years. The historical arguments have been reflected in the near past as Mendel’s concept has been diversified by new types of omics data. As an effect of the accumulation of omics data, a virtual gene concept forms, giving rise to genetical data science. The concept integrates genetical, functional, and molecular features of the Mendelian paradigm. I argue that the virtual gene concept should be deployed pragmatically. Indeed, the concept has already inspired a practical research program related to systems genetics. The program includes questions about functionality of structural and categorical gene variants, about regulation of gene expression, and about roles of epigenetic modifications. The methodology of the program includes bioinformatics, machine learning, and deep learning. Education, funding, careers, standards, benchmarks, and tools to monitor research progress should be provided to support the research program.

## 1. Introduction

I start by underlining the long-term significance of the Mendelian gene concept. Then, I argue that genetics is transformed into an interdisciplinary data science. Molecular genes are computationally modeled with various levels of biochemical detail and integrate various functional data. This computational modeling gives rise to a virtual gene concept in which genes are not physically existent, but appear to be so when being modeled at various levels of abstraction [1] as database records, programming objects, or elements of computer graphics. Finally, I argue that the virtual gene concept can shape a vibrant research program if applied pragmatically.

In his seminal 1866 paper [2], Gregor Mendel proposed a paradigm of intra-cellular, discrete, and particulate differentiating element mediating heredity (differirenden Zellelemente in German). Due to the limited experimental methodology available to him, the postulated particle of heredity was thought of in purely abstract terms. It was only later given the name we know today: *gene*. Mendel provided strong empirical evidence in support of his hypothesis. The evidence was in the form of brilliantly designed experiments on plant hybridizations. Moreover, Mendel analyzed the implications of the paradigm very well. His insights were profound, mathematical, and reached into the contemporary genomics era [1,3,4,5,6,7,8,9,10].

At the beginning of the 20th century, the gene continued to be seen as an abstract concept. However, later in the century, the reductionism [11] of molecular biology promoted a material—physicochemical—view of the gene. The material gene focused on the bio-organic chemistry of DNA, the genetic code, and the protein sequence defined in the order of exonic base pairs [7,8,9,10,12,13].

There is a danger that genetical data science could be an observational field drowning in data, particularly when practiced by the inexperienced. As a remedy, there is a need for research programs; that is, the clearly defined and well-reviewed sequences of theories. According to Lakatos [14], a scientific research program has a hard core of central theories that are regarded as certain and a soft shell of satellite falsifiable theories. There is also a group of practitioners who are interested in such theories or can apply them for practical purposes. The aim of this opinionated review is to argue for the support, funding, formalization of goals and achievements, benchmarking, and methodological development of the research program that is focused on the virtual gene concept.

## 2. Integrating Genomics Data Gives Rise to the Virtual Gene Concept

In the 21st century, data from the Human Genome Project (HGP) were integrated with data from functional genomics consortia. The functional genomics projects were designed to functionally characterize genomes of model species, humans, and other vertebrates. For example, a database of single-nucleotide polymorphisms (dbSNP) was developed. There were hundreds of genome-wide association studies (GWAS), or a database of expressed sequence tags (dbEST) [15], and a project for functional annotation of mammalian genomes (FANTOM). The first FANTOM catalogue [16] was based on sequencing full-length transcripts and revealed the existence of widespread antisense transcription [17] and many complex loci [18]. Moreover, FANTOM and RNA-seq [19] confirmed the existence of ubiquitous alternative splicing.

Presently, most functional genomic data are generated using next generation sequencing (NGS). For example, the FANTOM consortium developed a new technology for single-base resolution expression profiling called cap analysis of gene expression—CAGE [20]. Similar to dbEST, CAGE was applied to survey genome-wide transcriptional activity in human and mouse. The upgraded expression catalogue showed that transcriptional activity is spread widely throughout the genome, and that most human genes have distant enhancers [21], as well as multiple transcriptional start sites (TSSes). Other studies suggested that the TSSes likely evolved in correlation with splicing [22,23].

Additionally using NGS, a consortium developed that was dedicated to generating an Encyclopedia of DNA Elements—ENCODE [24]—and parallel encyclopedias for model species—modENCODE [25]. These encyclopedias characterized the patterns of DNA binding for dozens of transcription factors in humans, the fly, and the worm.

## 3. The Virtual Gene Concept Is Helpful in Genetics Education, in Computational Biology Research, and as a Focus of Data Integration

Students of genetics have learnt from undergraduate textbooks about Mendel’s paradigm using classic examples of the genes responsible for monogenic diseases, such as cystic fibrosis, or multiple genes controlling eye color or continuous traits such as height. Pedigree charts are shown to demonstrate different patterns of heredity, and basic information is provided about genetic mapping and molecular gene structure.

However, students today are also likely to have an early interactive encounter with the virtual gene concept using genome browsers. The three most popular genome browsers are: (1) European Bioinformatics Institute’s Ensembl; (2) a browser provided by the National Center for Biotechnology Information—NCBI; and (3) a browser provided by the University of California in Santa Cruz—UCSC.

Similarly, in computational genetical research, we increasingly interact with the virtual gene concept using bioinformatics tools. Both anti-reductionist and reductionist approaches can be used [1], but interactivity makes this classic methodological distinction less clear. For example, genome browsers pragmatically integrate different levels of abstraction, hyperlinking phenotypes or physical and genetic maps with sequence data. Note that a pragmatic tradition in philosophy suggests that bioinformaticians ought to focus on how well the variants of the gene concept work for practical problem solving, prediction, and action.

The pragmatism is evident in resources provided by the leading data integrator at the National Institutes for Health: NCBI [26]. NCBI’s gene database hyperlinks nucleotide sequences with a wide range of annotation databases, though mapping remains a challenge, being hard to standardize or describe statistically. A desire to pragmatically integrate as much data as possible appears as a priority, rather than ensuring that the mapping is completely non-redundant, always reproducible, or is statistically fully described.

In a second example, the UCSC Genome Browser [27,28,29,30,31] represents genes as pragmatic graphical objects that can be interactively repositioned or zoomed-in on. Each locus is annotated in a practical manner with genetic and functional data in parallel tracks. The genes are also hyperlinked to “Gene Views” containing nucleotide/protein sequences, to biomedical and biotechnological literature, gene expression data, or experimental protein structures.

Note that the trend towards data integration is accelerating [32,33,34,35,36,37,38,39]. Integrated, gene-related data may now include single-cell/spatial expression profiles [40,41,42], digital histopathology [43,44,45], or clinical imaging [46]. Indeed, data integration is now possible for omics datasets even at a single-cell level [47,48].

## 4. The Virtual Gene Concept Can Define a Practical Research Program

### 4.1. Methods of the Research Program

Traditionally, computational biology relied on purpose-developed bioinformatics methods [49,50,51]. Such bioinformatics tools are now well-described in well-edited textbooks and monographs; they are described either in a general context [49] or in more specific contexts, such as NGS informatics [51] or molecular evolution [52]. There are also numerous monographs devoted to specific technologies of genomics, such as the analysis of variability [53] or microarrays [50].

However, genetical data science now requires not only tools of bioinformatics but also of applied statistics, such as computer-age and large-scale statistical inference [54,55], techniques of statistical learning [56], or probabilistic modeling [57]. Moreover, algorithms of machine learning [58] and deep learning [59,60,61] have advantages over statistical models if associations between variables go beyond linearity and additive. For example, support vector machines, random forests, or dense neural networks detect non-linear or non-additive interactions between inputs. Convolutional neural networks—CNN—can recognize spatial associations between variables and non-additive effects. Recurrent neural networks can also be used to model spatial effects, for instance in the case of sequence data.

Generally speaking, if one can compile a set of functionally related nucleotide sequences, then they can be learnt as a computational model. The model can then be used for motif interpretation or prediction. This procedure is easy to apply for some of the in silico hypotheses generated by the virtual gene concept (in particular, those outlined below as (a), (b), and (c)). Deep learning offers new opportunities in this area as complex sequence motifs can be detected more flexibly than with statistical or machine learning. That is to say, deep learning robustly learns functional motifs in the whole sequence range covered by model inputs regardless of the positions of motifs in relation to each other. Moreover, sequence motifs can be flexibly combined with functional experimental data such as protein/DNA interactions, splicing events, post-transcriptional modifications, etc.

### 4.2. Goals of the Research Program

There are a number of research questions in which statistical data modeling, machine learning, and deep learning intersect non-trivially with the virtual gene concept. Such intersections suggest goals for the research program proposed here. The investigation of pseudo-genes was a successful example of such research [13,62,63,64,65,66,67,68,69,70,71,72,73,74]. In the future, a number of further in silico-testable hypotheses about the nature of genes could be included. Representative examples of such hypotheses are provided below.

(a)Alternative TSSes can drive contrasting patterns of gene expression [75,76,77]. This is not only because of the use of alternative promoters, but also because alternative 5′-UTRs might confer different mRNA stability, translational efficiency, or affect polymerase II (Pol II) pausing (Pol II pausing is increasingly recognized as having a regulatory role [78]). Note that alternative TSSes can switch during development [79,80], in response to stress [77], or in disease states [81]. However, it is not known how common or complete the switching is, or what all the biological functions are. The most interesting of such expression variants could have roles in cell-specific regulatory networks of somatic tissues, in development, or in cancer. Alternatively, co-expressed TSSes [82] may have evolved for regulatory robustness, as buffers against mutations, or simply to increase the transcriptional output of weak promoters.(b)Of course, alternative TSSes must produce alternatively structured transcripts. While many such transcripts have been previously sequenced and deposited in nucleotide sequence databases, they were typically attributed to alternative splicing rather than to alternative promoter usage. This fairly trivial deduction prompts a number of non-trivial questions. Crucially, do alternative TSSes result in neo-functionalized protein isoforms if the first exon is skipped? For example, truncated dominant negative members of protein complexes could bind a ligand or co-receptor but do not have enzymatic activity to propagate a signal. Alternatively, protein isoforms could differ in subcellular localizations if a signal peptide is affected; examples include *Arabidopsis* glutathione S-transferase F8 [77]. (Note that fusion transcripts [83], scrambled neighboring transcripts [84], or prematurely terminated transcripts [85] can give rise to functional protein variants, particularly in cancer. Pathological neo-functionalization is well-known for fusion transcripts resulting from chromosomal re-arrangements in cancer, e.g., constitutively active BCR-ABL1 fusion tyrosine kinase in chronic myeloid leukemia, which is successfully targeted using tyrosine kinase inhibitors [86]).(c)Are different varieties of non-coding transcripts generally functional genes? Or are they more commonly a type of biological noise that results from open chromatin in transcriptionally active chromosomal domains? There are already many known examples of functional antisense [87,88,89], functional expressed pseudo-genes, and functional long non-coding RNAs [67,69,71,73]. The quantification of genome-wide trends is needed, now.(d)Are enhancers and insulators typically associated with individual genes or rather with large but linear transcriptional domains? Are the transcriptional domains of universal significance, or do they only play a role in specific cell types? To what extent do the transcriptional domains correlate with the 3D organization of the genome? Early studies using a technique called optical reconstruction of chromatin architecture—ORCA—to image chromatin in embryos [90] suggest that transcriptionally active domains are sharply defined by borders of Polycomb-repressed DNA, but change with cell identity [91]. A deep learning model trained with ORCA data proved that 3D chromatin architecture strongly correlates with gene expression, and the effect is complex and diffuse, extending beyond direct contact of sequence-defined motifs such as promoters or enhancers [92]. (In fact, the contact of promoters with enhancers was not a good predictor of transcription [92]).(e)Can DNA sequences be usefully modeled on their own, or should DNA methylation, nucleosome occupancy, and histone modifications also typically be taken into account? For example, it is well known that several distinct types of histone modifications correlate positively with transcription [93,94]. However, there is no certainty if any of the histone modifications are early causative events. Rather, current knowledge suggests that pioneering transcription factors are primary causative agents for active transcription while the histone modifications are merely a later/downstream consequence of transcriptionally active chromatin [95,96,97,98]. On the other hand, DNA hyper-methylation and dense nucleosome occupancy in promoter regions appear to be early events of transcriptional silencing [99,100].

For each of the above hypotheses, there are intersections with further questions about genetic variability in populations, the impact on sequence alignment, or the calculation of distances/evolutionary rates between sequences. For example, one might ask about different kinds of polymorphisms, especially SNPs and small indels. Do the polymorphisms affect promoter usage, splicing, interactions between coding and non-coding transcripts, functions of enhancers/insulators, or the establishment of epigenetic marks? Moreover, should alternative TSSes be taken into account in sequence alignment or for the calculation of distances?

Determining which datasets will need integration in preparation for data mining to tackle such questions will, of course, depend on context. For some applications, alternative splice variants or promoters may not matter. For other projects, the focus will be precisely on the nuances of gene definition, or structure, or variability, or epigenetics, or on functional annotations.

### 4.3. Recent Examples

Deep learning has already been applied to predict alternative poly-adenylation [101,102], noncoding variants that interfere with splicing [103,104], gene regulatory networks [105,106,107,108], the expression of copy number variants [109], and in single-cells [110,111] or the targets of non-coding RNAs.

Table 1 and Table 2 list a few published examples of applications of deep learning related to the virtual gene concept. Table 1 is focused on simple model gene expression. Table 2 lists three examples focused on gene structure: promoter prediction or prediction of alternative splicing coordinated with polymerase II pausing. For each of the examples, the tables list references, the main result of the study, their biological interpretations, and the data inputs/outputs of the deep learning model.

Of course, Table 1 and Table 2 are intended to be illustrative rather than comprehensive. Indeed, it would be difficult to be comprehensive in such a dynamic field; important papers are published every month and any review will be outdated by the time of publication.

### 4.4. Potential Weaknesses of the Research Program

An obvious area for improvement in deep learning is model interpretability. Note that deep learning was developed as an enabling technology for industrial applications such as artificial vision and natural language processing. In such industrial applications, the predictive performance of models is prized over their interpretability. In scientific applications, however, the priorities are frequently exactly the opposite. Academic reviewers are likely to be as interested in the mechanisms of genetical phenomena as in prediction. Already, predictive black box models are less valued than *transparent* models from which functional sequence motifs—such as splice sites or transcription factor (TF) binding sites—can be extracted. Although several computational methods for enhancing interpretability in deep learning were proposed, successes are still mostly limited to the recovery of sequence motifs [112,113].

Moreover, interpretability itself may require a better theoretical framework [114] so that gradual improvements in interpretability can be quantified or benchmarked.

Note that if interpretability is of much higher value than prediction, traditional statistical data modeling using some flavor of multivariate statistics may be preferred in practice. This is because inference and learning theory is more developed in statistics than in computer science [115,116]. Moreover, visualizations are well developed to help in the diagnosis and interpretation of statistical models [117].

Of further note, both statistical and machine learning models may need adapting to specific genetic applications. For example, Teschendorff and Relton discussed adapting feature selection to the context of the analysis of DNA methylation data [118].

Hopefully, some universally applicable guidelines for variable selection and choice of algorithm will be developed with time. For example, in statistical data modeling, a useful rule of thumb may be to use only those variables that add to the predictive power of the model. Furthermore, generally speaking, automatic variable selection outperforms manual/human variable selection [119]. Common approaches to reduce the number of trainable parameters involve regularization steps (e.g., dropout, L1/L2 regularization, etc.) Note, however, that some gene-related variables may add predictive power to gene models [95] but could take away from interpretability. For example, I noticed that a chromatin-associated gene variable such as DNASE1/methylation signal or GC- and CpG-content had great predictive value for classifying genes as either housekeeping or tissue-specific [95]. However, the chromatin-associated variables had no explanatory value as the casual association for the breadth of expression was with the number of transcription factors binding a proximal promoter [96].

### 4.5. Related Research Programs

One can identify a number of successful research programs in genetical data science that are related but not focused on the concept of the gene itself. For example, systems genetics [120,121] focuses on the interpretation of phenotypes and has already yielded profound insights into widespread genetic pleiotropy [122,123]. Genomic prediction has revolutionized breeding and animal science over the last two decades. Network medicine aims to explain diseases [124,125].

Deep learning is currently making a great impact across all these related fields. Its applications have already been reviewed for general omics [126,127,128,129], gene function prediction [130,131], disease prediction [132], predicting the impact of genetic variation in genomics [129,133], predicting gene regulatory networks [133,134], regulatory genomics [135], sequence motifs of transcription factors and enhancers [133,134,136,137,138,139,140,141,142], variant calling and pathogenicity scores [143], precision medicine [144,145], pharmacogenomics [128], and even the prediction of CRISPR targets [146].

In this opinionated and forward-looking review, I argue for the formalization of a new research program in genetical data science. The program should be focused on the concept of the gene. Although related to fields mentioned in the two previous paragraphs, the proposed research program is markedly different. Crucially, the research program is a part of biology that is understood as a basic science. Its central question—gene concept—has long been a focus of theoreticians of biology and its philosophers [7,9,10,147].

### 4.6. The Need for Training, Funding, Benchmarking, and Monitoring of Progress

My goal was to define a research program named after the object of study: the virtual gene concept. Less experienced or younger researchers could benefit the most from such “manuals for doing research”. There can be no doubt that rich opportunities are available. However, success will always depend on the skills and drive of individual researchers. Data mining is a difficult and time-consuming art, which calls for a wide range of skills, patience, good judgment, and a perfectionist attitude. An understanding of the theoretical and philosophical aspects of gene concept will also be necessary. Ultimately, principal investigators—especially those who work individually rather than in consortia—will develop their own preferred way of practicing the data science. Postgraduate education, research funding, and a career structure in academia must be provided for this to be realistic. Methods developed in the field of virtual gene concept should be, of course, standardized and benchmarked. Ideally, methods could additionally be developed to quantitatively monitor the progress of research in the field.

## Figures and Tables

**Table 1 entropy-24-00017-t001:** Examples of deep learning applied to the problem of prediction of gene expression.

Reference	Main Result	Biological Interpretation of the Model	Data Inputs	Model Output	Other Points
Rajpurkar et al. [92].	Convolutional neural networks predict gene expression better than dense neural networks or a random forest. Blanking could reveal important motifs (for example, enhancers and silencers).	Chromatin architecture predicts gene expression. However, the effect is diffuse, extending beyond sequence-identifiable motifs such as promoters or enhancers.	Optical reconstruction of chromatin architecture (ORCA) of the Bithorax gene cluster in *Drosophila*. ORCA images were pre-processed to preserve structure but not viewing angle.	ON or OFF binary prediction of expression.	This was a remarkably innovative approach building on the strength of a remarkably novel dataset.
Zrimec et al. [148].	Up to 82% of the variation of transcript levels could be predicted from DNA sequences.	Both coding and cis-regulatory regions contribute to prediction of gene expression.	DNA sequences of proximal promoters *, plus 64 codon frequencies from coding regions, and eight mRNA stability variables.	Expression levels recoded as transcripts per million.	Motif interactions were key for the control of the dynamic range of gene expression.
Singh et al. [149].	A model derived from histone marks predicts expression better than traditional machine learning.	Histone marks correlate with expression, although it is unclear which marks are causative.	Histone marks from 56 different cell types [150] around TSSes in consecutive intervals.	Binary *high* or *low* gene expression level.	Complex interactions of chromatin features could be detected and visualized for intuitive interpretation.
Cuperus et al. [142].	A CNN trained on random 50 bp 5′-UTRs can predict the expression of a reporter gene from both artificial and native UTRs.	Alternative 5′-UTRs confer different mRNA stability or translational efficiency.	Nucleotide sequence of the 5′-UTR.	Scalar score for each UTR (proportional to protein expression).	Shorter UTRs did not work as well.

* One-hot encoding is a standard approach to sequence representation in deep learning.

**Table 2 entropy-24-00017-t002:** Examples of deep learning applied to structural annotation of virtual genes.

Reference	Main Result	Biological Interpretation of the Model	Data Inputs	Model Output	Other Points
Oubounyt et al. [151].	The prediction method improves performance over comparable approaches (fewer false positives). The improvement is attributed to a novel negative learning set.	Short eukaryote promoter sequences are sufficient to predict both TATA and non-TATA promoters in both human and mouse.	Genomic sequence from −249 to +50 bps relative to the TSS *.	A real-valued promoter score.	Impact of mutations on output scores was also studied (150 substitutions on the interval from –40 to +10 bps relative to the TSS).
Kelley et al. [152].	The method uses chromatin accessibility to predict gene promoters with high accuracy.	Promoters and transcription factor binding motifs could be predicted, but the method was developed to annotate point mutations.	DNase-seq mapping accessible genomic sites in 164 cell types (encoded as a binary vector of length 164). Plus, a DNA sequence of 600 bps *.	Probability value for chromatin accessibility.	Every mutation in the genome could be annotated with respect to its impact on chromatin accessibility.
Feng et al. [153].	A deep learning model can predict Pol II pausing events. (An attention layer and data integration provide good interpretability.)	The pausing events also provide insights into alternative splicing, TF binding sites, and epigenetic modifications.	200-50-bp DNA sequence * integrated with ChIP-seq and epigenetic data via an attention layer.	Probability value for Pol II pausing.	Strongest sequence determinants were typically −14 to 12 bp around the pausing sites. The model was relatively interpretable due to an attention mechanism analogous to DeepHINT [154].

* One-hot encoding is a standard approach to sequence representation in deep learning.

## Data Availability

The data is contained within the article.

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
