# Peer review of "Virtual Gene Concept and a Corresponding Pragmatic Research Program in Genetical Data Science"

_entropy, 2021, doi:10.3390/e24010017_

Round 1

Reviewer 1 Report

Dear author,

I read your article with title: Virtual gene concept and deep learning inspire a research program in genetical data science. I have to say that, although the topic is interesting, and you are certainly conversant with gene modelling, I struggle to fully grasp the main points of this work: this is certainly not a research article, but also not a systematic review of deep learning applied to genetics; is this an opinion paper ... something else? You should consider refocusing your work to make the objectives clearer.

English needs a little improvement, and I would kindly ask you to include line numbers, which would make the task of reviewers easier.

Finally, the paper could be much shorter without losing relevant content.

A few specific comments are listed below.

5. Deep learning is the newest data technology applied to investigate virtual genes
-----------------------------------------------------------------------------------------
- I suggest writing "computer-age" with hyphenation, to avoid ambiguity
- "Data modelling may be first used to identify most relevant explanatory variables for a given response variable. Then machine learning may be also applied.": it is not really clear to what the author refers. If this is to say that you can use classical data modelling / statistics to gain interpretable insights and learn something more about the phenomenon under study, I agree. If on the other hand you mean identifying the most relevant variables to be then fed to a machine learning method, there is the risk of "cherry picking", if validation of results is not properly designed (see for instance Tibshirani R, Hong W, Warnke R, Chu G, Staudt L, Wright G, et al. Immune signatures in follicular lymphoma. N Engl J Med,; and Hastie T, Tibshirani R, Friedman J. "Model assessment and selection". In The elements of statistical learning. New York: Springer; 2009. p. 219–60.). Additionally, automatic variable selection probably outperforms manual/human variable selection (e.g. Miotto, Riccardo, Li Li, Brian A. Kidd, and Joel T. Dudley. "Deep patient: an unsupervised representation to predict the future of patients from the electronic health records." Scientific reports 6, no. 1 (2016): 1-10.)
- "Support vector machines, random forests or dense neural networks detect non-linear or non-additive interactions between inputs. Convolutional neural networks — CNN — can recognize spatial associations between variables and non-additive effects.". The mentioned methods can of course also detect linear and additive relationships between variables, but are not limited to these and can effectively be used to go beyond linearity and additivity. As for spatial relationships, also RNN -recurrent neural networks- can be used to model them, for instance in the case of sequence data.
- the bit on variable selection (e.g. forward and backward stepwise selection etc.) seems to be out of place in this section on deep learning: the object of deep learning is precisely to learn automatically which variables to use and in which way, without previous variable selection steps. More common approaches to reduce the number of trainable parameters are regularization steps (e.g. dropout, L1/L2 etc.) or the use of specific architectures (e.g. CNN)
- about the interpretability of deep learning models, some efforts in this direction are done in the visualization of data (e.g. Wickham, Hadley, Dianne Cook, and Heike Hofmann. "Visualizing statistical models: Removing the blindfold." Statistical Analysis and Data Mining: The ASA Data Science Journal 8, no. 4 (2015): 203-225.)

Author Response

Reviewer 1.

>In the manuscript entitled Virtual gene concept and deep learning inspire a research program in >genetical data science, the author Lukasz Huminiecki briefly summarizes the historical evolution of >the gene concept and how the huge amount of information accumulated nowadays permit envisioning >the inception of a new research field that focuses on the complexities of “modern genes”. I find this >manuscript worth being published by Entropy. I only have three minor recommendations for the >author to consider while reviewing the manuscript for publication.

I would like to thank the referee for their positive review. I also appreciate the recommendation to publish my manuscript.

>Virtual is one of those adjectives with many nuances that can be interpreted differently by different >people. Also, it is a word that it is not uncommon in scientific literature. Because of these two points, >it is very possible that in a piece talking about virtual entities we encountered instances of the term >virtual and/or variations of the term being used with different purposes, which can be very confusing >for the reader. In the first paragraph of section 6, the author uses “virtually” and the “virtual gene >concept” to mean slightly different things. To minimize this issue, I would encourage the author to >offer a “concrete” definition of the virtual gene concept as early as possible and avoid using the term >with slightly different meanings all throughout the manuscript. An alternative would be to find a less >potentially contentious alternative to virtual, but I am afraid that I have not found a right >recommendation. I feel that “extended gene concept” would capture only partly what the author wants >to convey with the expression “virtual gene concept”.

I fully agree with the suggestion of the referee to simplify how I use the word virtual. The article now only uses the word virtual in the context of the phrase virtual gene concept.

>If I understand correctly, the author pushes in this manuscript for the formalization of a research >field/program based on the virtual gene concept, that would be also named “virtual gene concept”. Is >this correct? I find this a little bit confusing, specially when the author defends the usefulness of this >new field by comparing it with other potentially related but distinctive fields that are not name after >the subject of study, e.g., systems genetics, genomic prediction, or regulatory genomics. Yet again, I >have no better alternatives.

Indeed, the main objective is to argue for the support, funding, formalization of goals and achievements, benchmarking, and methodological development within the research program focused on the virtual gene concept.

>Lastly, I find the titles of sections 4 and 6 conceptually similar, and therefore redundant.

The manuscript is now reo-organized to make its objectives clearer. The relevant sections have different order and different titles. Please, let me know if you find any sections still redundant.

I also changed the title to underline a focus on the research program. The title is now: Virtual gene concept and a corresponding pragmatic research program in genetical data science.

Reviewer 2 Report

In the manuscript entitled Virtual gene concept and deep learning inspire a research program in genetical data science, the author Lukasz Huminiecki briefly summarizes the historical evolution of the gene concept and how the huge amount of information accumulated nowadays permit envisioning the inception of a new research field that focusses on the complexities of “modern genes”. I find this manuscript worth being published by Entropy. I only have three minor recommendations for the author to consider while reviewing the manuscript for publication.

Virtual is one of those adjectives with many nuances that can be interpreted differently by different people. Also, it is a word that it is not uncommon in scientific literature. Because of these two points, it is very possible that in a piece talking about virtual entities we encountered instances of the term virtual and/or variations of the term being used with different purposes, which can be very confusing for the reader. In the first paragraph of section 6, the author uses “virtually” and the “virtual gene concept” to mean slightly different things. To minimize this issue, I would encourage the author to offer a “concrete” definition of the virtual gene concept as early as possible and avoid using the term with slightly different meanings all throughout the manuscript. An alternative would be to find a less potentially contentious alternative to virtual, but I am afraid that I have not found a right recommendation. I feel that “extended gene concept” would capture only partly what the author wants to convey with the expression “virtual gene concept”.

If I understand correctly, the author pushes in this manuscript for the formalization of a research field/program based on the virtual gene concept, that would be also named “virtual gene concept”. Is this correct? I find this a little bit confusing, specially when the author defends the usefulness of this new field by comparing it with other potentially related but distinctive fields that are not name after the subject of study, e.g., systems genetics, genomic prediction, or regulatory genomics. Yet again, I have no better alternatives.

Lastly, I find the titles of sections 4 and 6 conceptually similar, and therefore redundant.

Author Response

Reviewer 2.

>I read your article with title: Virtual gene concept and deep learning inspire a research program in >genetical data science. I have to say that, although the topic is interesting, and you are certainly >conversant with gene modelling, I struggle to fully grasp the main points of this work: this is certainly >not a research article, but also not a systematic review of deep learning applied to genetics; is this an >opinion paper ... something else? You should consider refocusing your work to make the objectives >clearer.

I indented to be opinionated and forward-looking in this review. This is stated twice in the new text. The main objective is to argue for the support, funding, formalization of goals and achievements, benchmarking, and methodological development of the research program focused on the virtual gene concept.

>English needs a little improvement, and I would kindly ask you to include line numbers, which would >make the task of reviewers easier.

Line numbers are now added.

>Finally, the paper could be much shorter without losing relevant content.

The manuscript is shortened, and re-organized it to make its main objective and argument clearer. In particular, pages 7 – 14 (lines 128 – 317) now constitute one high-level section that discusses various features of the proposed research program. This new section discusses program’s methods, goals, examples, potential weaknesses, related research programs, as well as practical aspects such as need for training and funding.

If the reviewer recommends deleting further text, please, specify exactly which paragraphs should be removed.

>A few specific comments are listed below.
>5. Deep learning is the newest data technology applied to investigate virtual genes
>-----------------------------------------------------------------------------------------
>- I suggest writing "computer-age" with hyphenation, to avoid ambiguity

I use the phrase "computer-age" only once. It is, now, spelled with hyphenation.

>- "Data modelling may be first used to identify most relevant explanatory variables for a given >response variable. Then machine learning may be also applied.": it is not really clear to what the >author refers. If this is to say that you can use classical data modelling / statistics to gain interpretable >insights and learn something more about the phenomenon under study, I agree. If on the other hand >you mean identifying the most relevant variables to be then fed to a machine learning method, there is >the risk of "cherry picking", if validation of results is not properly designed (see for instance >Tibshirani R, Hong W, Warnke R, Chu G, Staudt L, Wright G, et al. Immune signatures in follicular >lymphoma. N Engl J Med,; and Hastie T, Tibshirani R, Friedman J. "Model assessment and >selection". In The elements of statistical learning. New York: Springer; 2009. p. 219–60.). >Additionally, automatic variable selection probably outperforms manual/human variable selection >(e.g. Miotto, Riccardo, Li Li, Brian A. Kidd, and Joel T. Dudley. "Deep patient: an unsupervised >representation to predict the future of patients from the electronic health records." Scientific reports 6, >no. 1 (2016): 1-10.)>- "Support vector machines, random forests or dense neural networks detect non-linear or non->additive interactions between inputs. Convolutional neural networks — CNN — can recognize spatial >associations between variables and non-additive effects.". The mentioned methods can of course also >detect linear and additive relationships between variables, but are not limited to these and can >effectively be used to go beyond linearity and additivity. As for spatial relationships, also RNN ->recurrent neural networks- can be used to model them, for instance in the case of sequence data.
>- the bit on variable selection (e.g. forward and backward stepwise selection etc.) seems to be out of >place in this section on deep learning: the object of deep learning is precisely to learn automatically >which variables to use and in which way, without previous variable selection steps. More common >approaches to reduce the number of trainable parameters are regularization steps (e.g. dropout, L1/L2 >etc.) or the use of specific architectures (e.g. CNN)
>- about the interpretability of deep learning models, some efforts in this direction are done in the >visualization of data (e.g. Wickham, Hadley, Dianne Cook, and Heike Hofmann. "Visualizing >statistical models: Removing the blindfold." Statistical Analysis and Data Mining: The ASA Data >Science Journal 8, no. 4 (2015): 203-225.)

The above statistical material is now in the section entitled Methods of the research program. Suggestions of the referee with regards to references were mostly followed.
